# Hybrid perovskite light emitting diodes under intense electrical excitation

Hoyeon Kim [1], Lianfeng Zhao [2], Jared S. Price[1], Alex J. Grede [1], Kwangdong Roh[2], Alyssa N. Brigeman[1], Mike Lopez[1], Barry P. Rand[2,3] & Noel C. Giebink [1]

Hybrid perovskite semiconductors represent a promising platform for color-tunable light emitting diodes (LEDs) and lasers; however, the behavior of these materials under the intense electrical excitation required for electrically-pumped lasing remains unexplored. Here, we investigate methylammonium lead iodide-based perovskite LEDs under short pulsed drive at current densities up to 620 A cm$^{-2}$. At low current density ($J < 10$ A cm$^{-2}$), we find that the external quantum efficiency (EQE) depends strongly on the time-averaged history of the pulse train and show that this curiosity is associated with slow ion movement that changes the internal field distribution and trap density in the device. The impact of ions is less pronounced in the high current density regime ($J > 10$ A cm$^{-2}$), where EQE roll-off is dominated by a combination of Joule heating and charge imbalance yet shows no evidence of Auger loss, suggesting that operation at kA cm$^{-2}$ current densities relevant for a laser diode should be within reach.

[1] Department of Electrical Engineering, The Pennsylvania State University, University Park, PA 16802, USA. [2] Department of Electrical Engineering, Princeton University, Princeton, NJ 08544, USA. [3] Andlinger Center for Energy and the Environment, Princeton University, Princeton, NJ 08544, USA. Correspondence and requests for materials should be addressed to N.C.G. (email: ncg2@psu.edu)

H ybrid organic–inorganic halide perovskite semiconductors are presently being explored for application in light emitting diodes (LEDs) and lasers because they combine high color purity with a broadly tunable bandgap and attractive gain characteristics[1–4]. In particular, perovskites may provide a route to achieve the long-standing goal of a solution-processed laser diode, surpassing organic semiconductors on the basis of their higher charge carrier mobility and lack of exciton annihilation processes[5]. Following numerous optically pumped perovskite laser demonstrations[6–10] that have now reached continuous-wave operation[11], the next major milestone on the path to a diode laser is stimulated emission under intense electrical excitation. Perovskite LEDs are the most well-established platform for electrical pumping and have demonstrated external quantum efficiencies (EQEs) over 20% at low current densities ($J < 1\,A\,cm^{-2}$)[12]; however, the high current density regime relevant for lasing ($J > 100\,A\,cm^{-2}$) remains relatively unexplored.

Here, we investigate the performance of methylammonium lead iodide (MAPbI$_3$)-based perovskite LEDs at short pulsed current densities up to $620\,A\,cm^{-2}$. At current densities less than $10\,A\,cm^{-2}$, we find that the EQE depends strongly on the time-averaged history of the pulse train and can change dramatically depending on the duty cycle or background bias, reaching a peak EQE of approximately 13.5%, which is well above that obtained under direct current (DC) operation. This behavior is associated with ion movement on a millisecond to second timescale that changes the internal field distribution and trap density in the perovskite layer[13–19], thereby affecting charge balance and non-radiative recombination losses, respectively. The impact of ions is less pronounced in the high current density regime ($J > 10\,A\,cm^{-2}$), where efficiency roll-off is dominated by a combination of Joule heating and charge imbalance, yet shows no evidence of Auger loss even at current densities exceeding 150 $A\,cm^{-2}$. These results demonstrate that the efficiency of perovskite LEDs depends strongly on the manner in which they are driven and indicate that operation at $kA\,cm^{-2}$ current densities relevant for electrically pumped lasing should be within reach.

## Results

**DC versus pulsed operation.** Figure 1a illustrates the LED device architecture, which consists of a 150 nm-thick indium-tin-oxide (ITO) anode on a glass substrate followed by a 25 nm-thick poly [N,N′-bis(4-butyl-phenyl)-N,N′-bis(phenyl)-benzidine (poly-TPD) hole transport layer, a 70 nm-thick MAPbI$_3$ emissive layer mixed with BAI (n-butylammonium iodide) in a 100:20 molar ratio, and a 40 nm-thick 2,2′,2″-(1,3,5-benzinetriyl)-tris(1-phenyl-1-H-benzimidazole) (TPBi) electron transport layer capped by a LiF (1.2 nm)/Al (100 nm) cathode[3]. The circular device active area shown in Fig. 1b is defined by patterning small holes with diameter ranging from 50 to 200 μm in a 150 nm-thick layer of insulating SiO$_2$ deposited on the anode, similar to previous work on high current organic LEDs[20]. The electroluminescence (EL) image in Fig. 1c at $J = 110\,A\,cm^{-2}$ confirms that the LED is spatially uniform and does not exhibit hot spots or edge effects.

Figure 2a shows the current density–voltage–light (JVL) characteristics measured for a single device under DC and pulsed conditions defined by the pulse width ($t_{on}$), time between pulses ($t_{off}$), and background bias ($V_{bias}$) as depicted in the inset. Similar to previous reports, hysteresis is evident in the forward and backward DC sweeps of both the current and light[3,21,22]. By contrast, the pulsed drive JVL data ($t_{on} = 15\,μs$, $t_{off} = 2\,ms$) exhibit virtually no hysteresis but depend strongly on the background bias, with both current and light increasing with $V_{bias}$ in the range 0 to 3 V. This in turn leads to pulsed EQE curves in Fig. 2b that increase with $V_{bias}$ and reach a substantially

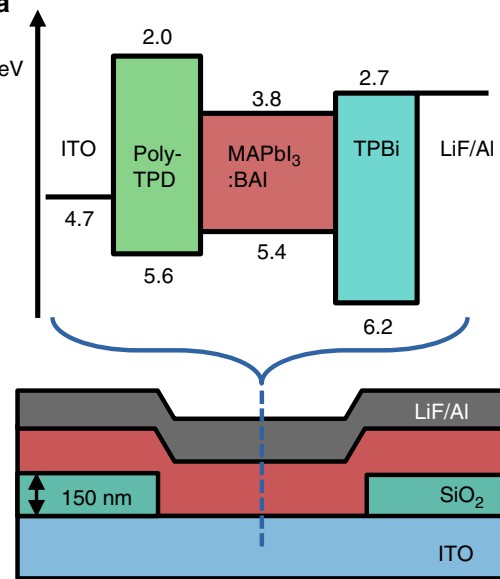

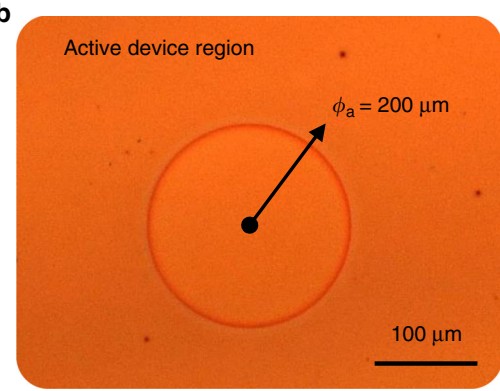

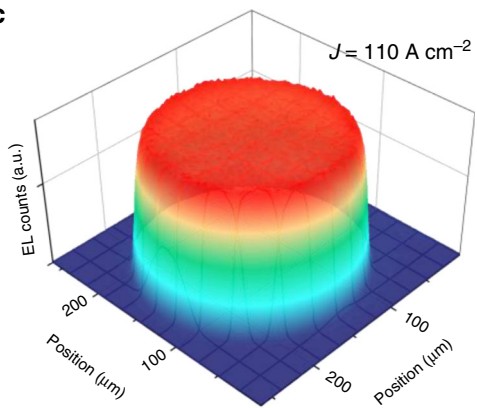

**Fig. 1** Device architecture. **a** Energy level diagram and schematic of the device architecture, which consists of ITO (150 nm)/poly-TPD (25 nm)/ MAPbI$_3$:BAI (100:20, 70 nm)/TPBi (40 nm)/LiF (1.2 nm)/Al (100 nm). The active area is defined by patterning openings in the insulating SiO$_2$ (150 nm) layer shown in the bottom graphic. **b** Optical microscope image of a typical 200 μm diameter device. **c** Electroluminescence intensity profile recorded with the microscope camera, demonstrating uniform illumination even at a pulsed current density of 110 $A\,cm^{-2}$

higher EQE peak of 13.5% than that obtained under DC drive (approximately 10%).

Interestingly, the EQE also depends on the pulse duty cycle, particularly with no background bias where, for example, the

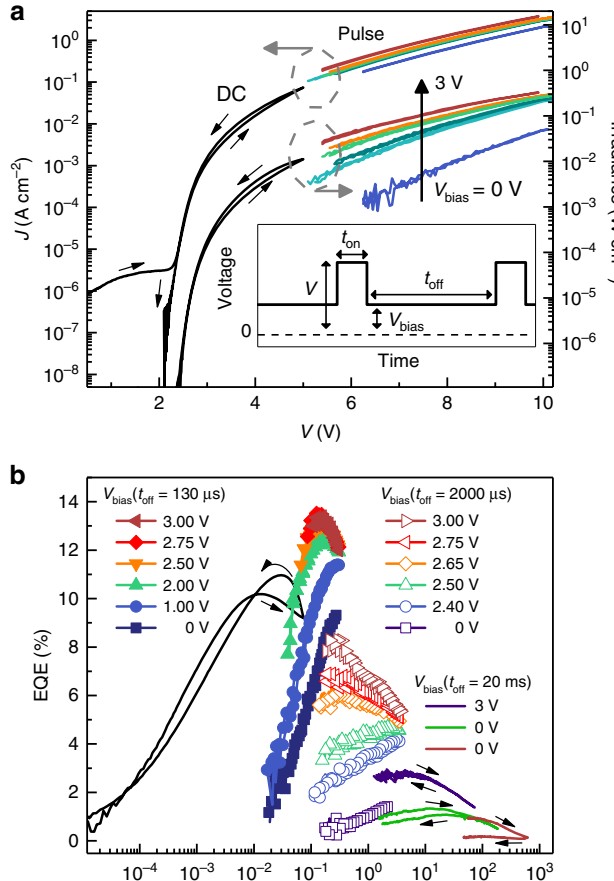

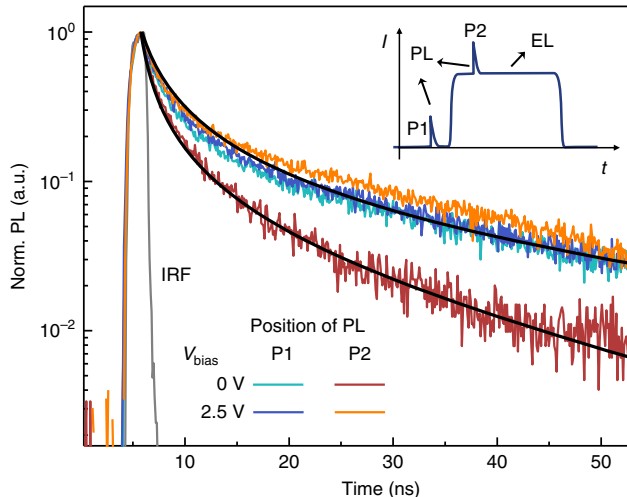

**Fig. 3** Transient photoluminescence during voltage pulses. Transient photoluminescence decays acquired from an optical pump pulse ($\lambda = 355$ nm, 20 ps pulse width, 0.2 µJ cm$^{-2}$ fluence) incident 1 µs before (P1) and after (P2) the start of a 0.05 A cm$^{-2}$ electrical pulse with different background bias levels as depicted in the inset diagram. Solid black lines are fits to the ABC rate equation described in the text. The instrument response function (IRF) is depicted by the gray line with a temporal resolution of 0.5 ns

**Fig. 2** Operation under DC and pulsed drive. **a** Current density (left-hand axis) and irradiance (right-hand axis) versus voltage for a 200 µm diameter device operated in DC (black lines) and pulsed (colored lines) drive; small black arrows indicate the direction of the voltage sweep. Hysteresis is imperceptible in the pulsed data. The pulse width and time between pulses illustrated schematically in the inset are $t_{on} = 70$ µs and $t_{off} = 2000$ µs, respectively. Increasing the background bias ($V_{bias}$) significantly increases the electroluminescence intensity as indicated by the vertical black arrow. **b** External quantum efficiency of the same device under DC and pulsed drive ($t_{on} = 70$ µs, 15 µs, and 2 µs for filled symbols, void symbols, and colored lines, respectively) with varying $V_{bias}$ and $t_{off}$. The highest current density trace (brown line) was achieved with a 50 µm diameter device that degraded during the reverse sweep

EQE during a $t_{on} = 10$ µs pulse increases by an order of magnitude at $J = 0.2$ A cm$^{-2}$ upon reducing $t_{off}$ from 2 ms to 130 µs; the functional dependence on $t_{off}$ is detailed in Supplementary Figure 1. Note that background bias not only changes the EQE magnitude, but also qualitatively changes its dependence on current density as exemplified by the open symbols in Fig. 2b, where EQE increases with $J$ when $V_{bias} = 0$ V but decreases when $V_{bias}$ exceeds 2.6 V. A practical consequence of this is that the current density at which the EQE reaches its peak can vary dramatically depending on duty cycle and background bias.

All of these effects become less pronounced in the high current density regime ($J > 10$ A cm$^{-2}$), where the EQE curves decrease monotonically and begin to converge irrespective of background bias (Fig. 2b). At a pulse width of 2 µs (near the minimum set by the parasitic capacitance of our device architecture), an EQE of approximately 1% is reliably achieved at $J = 200$ A cm$^{-2}$. Higher currents up to $J = 620$ A cm$^{-2}$ (limited by the driving circuit) can

be reached without catastrophic device failure; however, this leads to irreversible device degradation as evident from the return sweep of the brown curve in Fig. 2b. In general, we find that degradation is insignificant over the timescale of our measurements for current densities below 100 A cm$^{-2}$; Supplementary Figure 2 quantifies the rate of degradation observed at higher current densities. Taken together, the data in Fig. 2 clearly demonstrate that the performance of these perovskite LEDs is history dependent in the low current regime current but is governed by more general factors at high current. The following sections investigate each regime in more detail.

**Low current operation.** The impact of background bias on EQE can be understood in part from an associated change in carrier lifetime shown in Fig. 3. There, the photoluminescence (PL) decay of a weak optical excitation pulse (20 ps pulse width, $\lambda_{ex} = 410$ nm, 0.2 µJ cm$^{-2}$ fluence) is monitored just before, and just after, the onset of a voltage pulse with fixed current density, $J = 0.05$ A cm$^{-2}$. When $V_{bias} = 0$ V, the PL decay during the voltage pulse is quenched relative to that before the pulse. Adding a 2.5 V background bias eliminates PL quenching during the pulse, yet has no effect on the pre-pulse decay. Fitting these decay curves with a simple ABC rate model[23] indicates that the background bias decreases the trap-related A coefficient from $(4.0 \pm 0.4) \times 10^7$ s$^{-1}$ to $(1.5 \pm 0.3) \times 10^7$ s$^{-1}$. Interestingly, the bimolecular B coefficient in these fits also decreases by roughly a factor of 2, which may be related to a different internal field distribution (i.e., effectively lowering the encounter probability of photogenerated electrons and holes by sweeping them apart) created within the device as discussed below.

In addition to background electrical bias, Fig. 4a shows that the pulsed EQE also increases with optical bias, which is delivered here by a $\lambda = 660$ nm laser (scattered light is subtracted out in the baseline of the pulsed electroluminescence signal). The EQE improvement saturates above a background illumination intensity of approximately 10 mW cm$^{-2}$ and is reminiscent of the photoinduced brightening of MAPbI$_3$ PL previously reported by

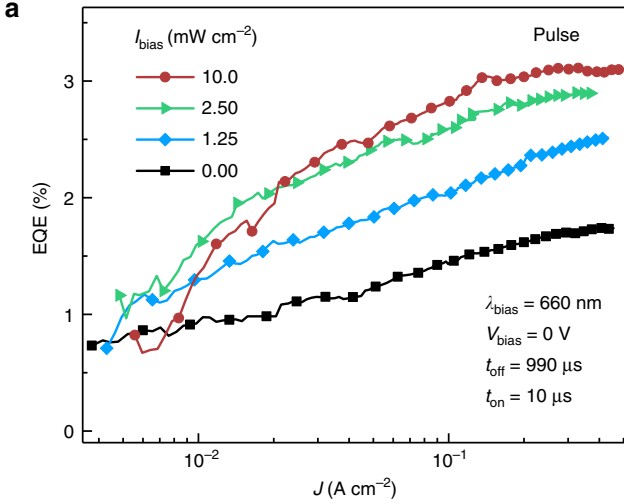

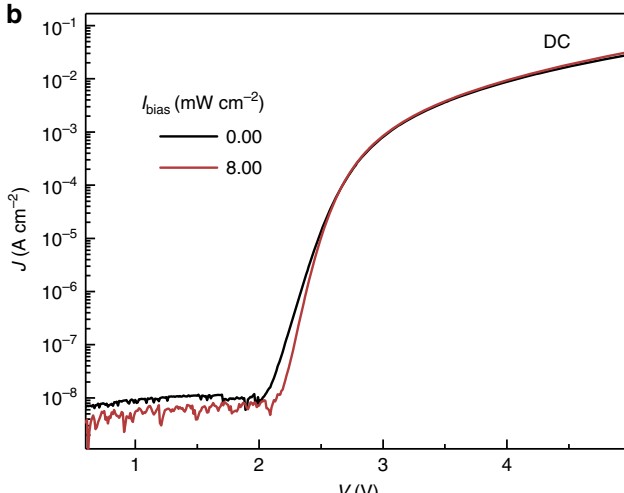

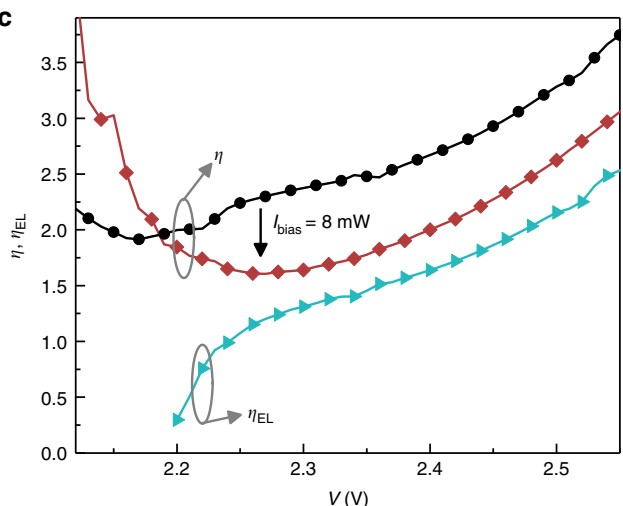

**Fig. 4** Effects of background illumination. **a** Pulsed external quantum efficiency recorded under varying levels of background illumination provided by a $\lambda = 660$ nm laser. **b** Under illumination, the slope of the DC current–voltage characteristic increases in the diode-like exponential region. **c** Ideality factor calculated for both the DC current ($\eta$) and electroluminescence ($\eta_{EL}$) as described in the text. Background illumination causes $\eta$ to decrease toward $\eta_{EL}$, reflecting a reduction in trap-mediated recombination

DeQuilettes et al.,[24] who attributed it to a reduction in ion-related trap states.

The illumination-induced change in current–voltage relationship displayed in Fig. 4b suggests that our EQE improvement with optical bias is associated with a similar reduction in trap-mediated recombination. In Fig. 4b, background illumination increases the slope of the exponential region in the JV curve, causing the associated ideality factor, $\eta = \left(\frac{kT}{q}\frac{\partial \ln J}{\partial V}\right)^{-1}$, calculated in Fig. 4c to fall from $\eta \approx 2.2$ in the dark to $\eta \approx 1.6$ under illumination. This difference suggests that the diode current in the device shifts from being dominated by trap-mediated recombination to include more (radiative) bimolecular recombination, for which $\eta$ is expected to be closer to unity as reflected in the electroluminescence ideality factor, $\eta_{EL} = \left(\frac{kT}{q}\frac{\partial \ln L}{\partial V}\right)^{-1}$, shown by the cyan curve[25].

Apart from changes in recombination mechanism, it is also clear from electroabsorption (EA) measurements shown in Fig. 5 that $V_{bias}$ non-trivially changes the internal electrostatic field within the device. Figure 5a shows the EA spectrum of a typical device obtained by measuring the reflectivity change ($-\Delta R/R$) in a probe beam caused by a dither superimposed on the bias voltage. The EA lineshape differs somewhat from that reported for MAPbI$_3$ previously[26], possibly due to the nanoscale grain size of our perovskite film[3], but is nonetheless confirmed to be a quadratic Stark shift (i.e., $\propto \chi^{(3)}(\omega, 0, 0)F^2$, where $F$ is the local electrostatic field in the perovskite layer)[27] based on its quadratic field dependence at high modulation frequency in Supplementary Figure 3. The square root of the EA signal therefore provides a direct measure of the electrostatic field in the perovskite layer and can be used to monitor changes that may occur due to, e.g., ionic charge redistribution over time.

Figure 5b shows the time-dependent EA signal[28] obtained at $\lambda = 730$ nm in response to a 10 ms period square wave oscillating between $-2$ V and $+2$ V (red trace). In this case, the EA signal (black trace) largely mirrors the applied voltage, indicating that the internal field in the perovskite follows accordingly. However, when the period is increased to 2 s in Fig. 5c, the EA signal decays strongly in both phases of the square wave, indicating that the applied field is being screened internally within the perovskite layer on an approximately 0.5 s timescale. This screening behavior and timescale are both consistent with that expected for ionic processes in MAPbI$_3$.

**High current operation**. The convergence of EQE curves in Fig. 2b at high current suggests that the hysteretic effects discussed in the previous section become dominated by more general factors that govern the EQE roll-off. Chief among these is Joule heating, which becomes significant at high current density despite our efforts to mitigate it through short pulsed drive and small device area. The current transients shown in Fig. 6a hint at this problem, evolving from the steady-state plateau maintained during low-voltage pulses to steady growth during high-voltage pulses. The latter regime is clearly detrimental to the EQE, where it leads to a corresponding decrease in EL intensity within the pulse duration as shown in Fig. 6b.

More direct evidence for heating comes from the peak shift and high energy spectral broadening of the pulsed EL spectra (red curves) presented in Fig. 6c. Both changes are expected for increasing temperature and are observed in control devices heated on a hot plate under low current density drive which are also shown in Fig. 6c for comparison (blue curves). Based on these reference data, we estimate that the temperature of the perovskite layer exceeds 345 K by the end of the 2 μs pulse when $J = 203$ A cm$^{-2}$. Though we acknowledge that this estimate could be complicated by spectral changes from unrelated phenomena such

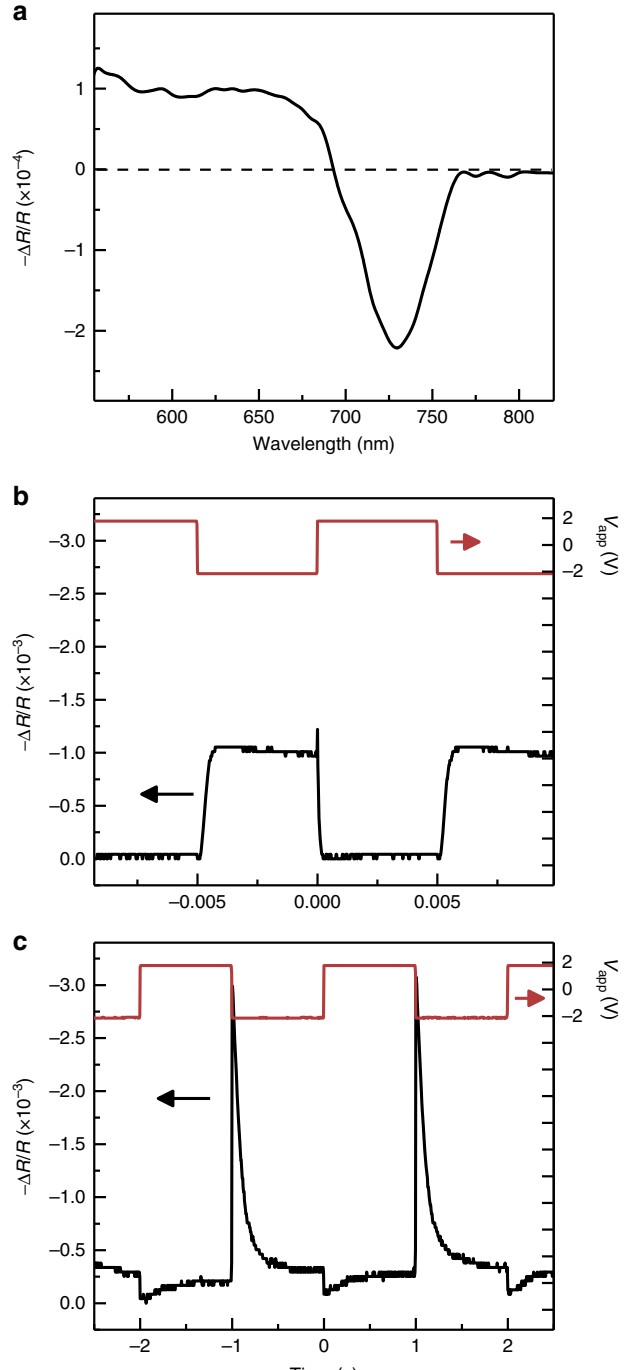

**Fig. 5** Electroabsorption probe of internal electric field. **a** Electroabsorption (EA) spectrum acquired in differential reflectivity from a device depleted in reverse bias at $-2$ V with a 0.8 V peak-to-peak sinusoidal dither at a modulation frequency of 389 Hz. **b** Transient EA signal resulting from a $-2$ to 2 V square wave signal with a 10 ms period. **c** Increasing the square wave period to 2 s leads to a decay of the EA signal during each half cycle that indicates screening of the internal electric field in the perovskite layer

as a shift in recombination zone discussed below, it agrees with finite element thermal modeling shown in Fig. 6d. There, the onset of non-negligible Joule heating occurs at $J\sim10$ A cm$^{-2}$ for 2 µs pulses and the transient temperature rise of the perovskite layer shown in the inset mirrors the functional form of the EL intensity decay reproduced from Fig. 6b, supporting the connection between the two.

Beyond the issue of heating, it is also clear that some level of carrier leakage and charge imbalance contribute to the EQE roll-off at high current density based on the transport layer emission at $\lambda\sim420$ nm recorded in Fig. 7. This emission emerges above $J\sim50$ A cm$^{-2}$ and strongly resembles poly-TPD photoluminescence (overlaid for comparison), indicating the presence of electron leakage into (and subsequent EL from) the hole transport layer. Hole leakage into the TPBi electron transport layer may also occur; however, any TPBi emission would likely go undetected due to strong absorption by the perovskite layer as illustrated in the inset diagram.

Figure 8 examines the possible role of Auger loss in a PL-on-EL experiment similar to that in Fig. 3, but carried out at higher current density and shorter pulse width. Figure 8a shows the timing sequence of the experiment, with the PL pulse arriving approximately 250 ns before or after the beginning of the EL pulse to minimize subsequent Joule heating per Fig. 6a, b. Any time-average Joule heating over the course of many pulses is ruled out based on the near-identical pre-pulse PL decay for all of the pulsed current densities shown in Fig. 8b, as an increase in temperature would be reflected by an acceleration of the decay rate[23]. Perhaps surprisingly, however, the in-pulse PL decays are also largely unaffected by the electrical excitation up to $J\sim156$ A cm$^{-2}$ (the damage threshold in this particular measurement). This observation contrasts with the increased decay rate that would be expected if Auger recombination (due to the large electrically-generated carrier density) or any other current density-dependent quenching mechanism were significant.

## Discussion

Taken together, the data above point to a picture in which perovskite LED operation at low currents is heavily influenced by the distribution of ionic charge, which affects both the trap density and internal field distribution in the device. Because the motion of ions—likely dominated by movement of iodide vacancies and interstitials[15,29] (which may have an intrinsic concentration greater than 0.4% at room temperature)[30] in the present case—takes place on a relatively long (millisecond to second) timescale, their configuration is essentially frozen during a short microsecond voltage pulse. Thus, it is the time-averaged history of the LED under pulsed operation (depending on factors such as $V_{bias}$ or average value of the duty cycle) that sets the internal field and trap distribution that electrons and holes injected on a microsecond timescale must contend with.

Figure 9 illustrates this model, which is similar in essence to that from ref. [15]. At zero bias, the built-in potential due to the difference in contact work function causes ions to drift toward the perovskite layer interfaces, screening the field in the bulk as shown in Fig. 9a. Under a sudden voltage pulse, the band diagram tilts accordingly but the ions remain frozen in place (Fig. 9b). The potential created by the ion distribution concentrates electrons and holes primarily at the perovskite layer interfaces, maximizing recombination where the ionic trap density is highest and thereby leading to large non-radiative loss. In contrast, applying an appropriate time-averaged forward bias (e.g., from $V_{bias}$ or the average duty cycle) reverses the equilibrium ion migration, causing the iodide vacancies to be refilled by the interstitials they leave behind (Fig. 9c). This reduces both the ionic trap concentration and its associated field distribution that confines electrons and holes near the transport layer interfaces, enabling radiative bimolecular recombination to occur more uniformly and efficiently throughout the perovskite bulk (Fig. 9d). Evidence of a such a shift in recombination zone follows from a slight red shift in the EL spectrum with increasing background bias (see Supplementary Figure 4), which is consistent with a

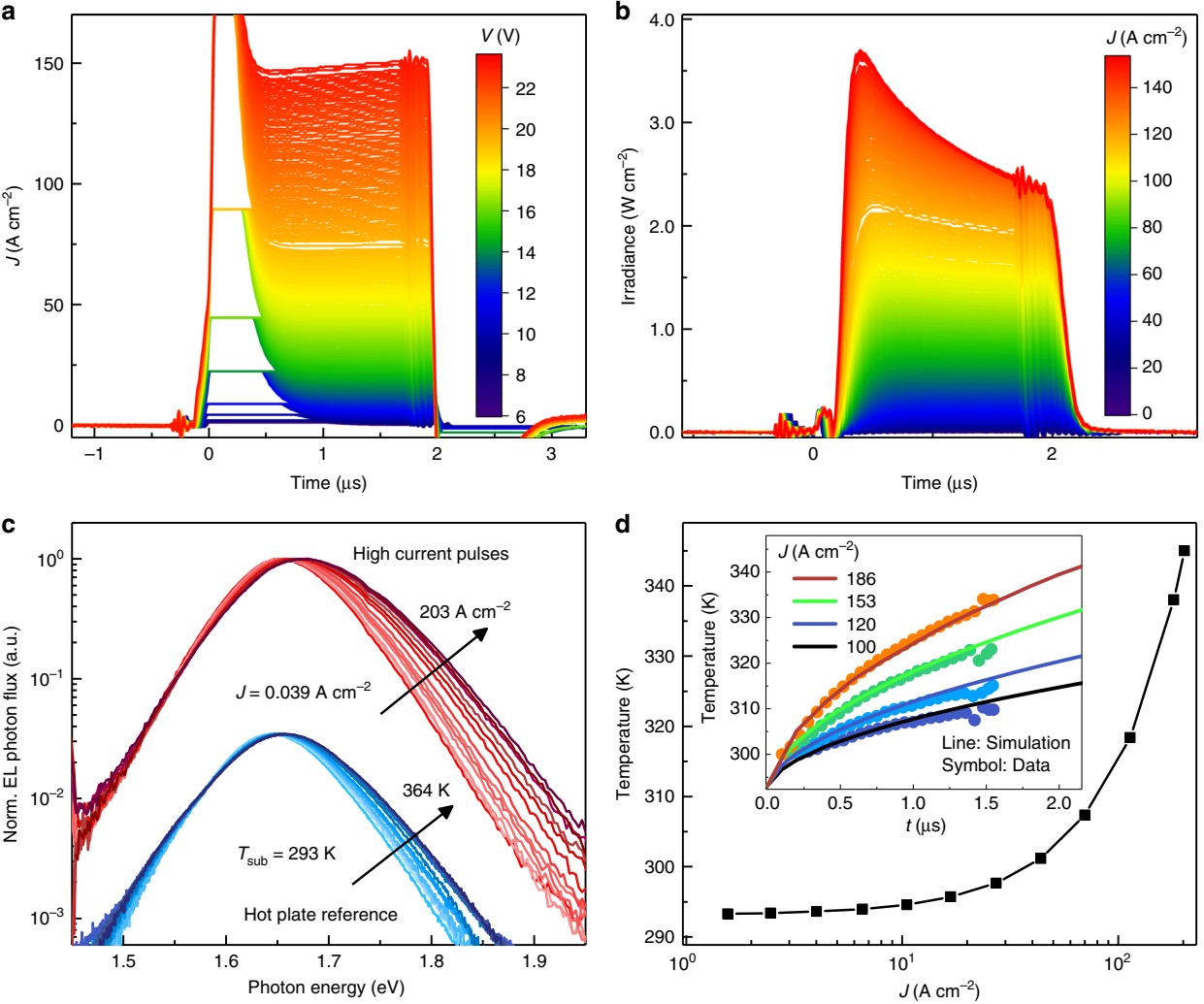

**Fig. 6** Joule heating at high current density. Transient **a** current density and **b** irradiance at different pulsed voltages. The gradual increase in current and decrease in EL during the higher voltage pulses is suggestive of Joule heating. **c** The EL spectra recorded from the same pulses (upper red data set) demonstrates a peak shift and high energy broadening that are similar to a device operated at low current ($J = 0.001$ A cm$^{-2}$ DC) while being heated on a hot plate (lower blue data set). **d** Finite element simulation of the average temperature in the perovskite layer at the end of a 2 µs pulse with varying current density. The inset shows the transient temperature rise simulated for several current densities (solid lines). The corresponding EL decays from **b** are overlaid (inverted and scaled by a single, constant factor) to highlight the connection between the two

recombination zone that moves from the TPBi interface toward the bulk of the perovskite layer.

This model also rationalizes the PL decay changes in Fig. 3, where non-radiative recombination is increased during the voltage pulse with no background bias because photogenerated carriers are swept by the internal field to the perovskite interfaces where the ionic defect concentrations are highest (Fig. 9b). The same traps have less of an influence on the pre-pulse transient because the internal field is screened and photogenerated carriers therefore recombine mainly in the perovskite bulk (Fig. 9a). The situation is essentially the same (i.e., photogenerated carriers recombine in the perovskite layer bulk) for the pre-pulse transient with background bias applied since the internal field is similarly suppressed (Fig. 9c). During the voltage pulse, however, carriers that are swept to the perovskite layer edges now have fewer ionic defects with which to undergo non-radiative recombination (Fig. 9d).

Previous reports have demonstrated that background optical bias has a similar effect in reducing the iodide defect concentration in MAPbI$_3$[24], which is consistent with the EQE improvement and decrease in trap-mediated recombination current observed in Fig. 4. Beyond simply changing the balance of radiative versus non-radiative recombination, it seems likely that optical and/or electrical background bias also affects the charge balance of the device due to the variation in internal field that accompanies the changing ionic charge distribution. Significant changes in the EQE-current density functional relationship (Fig. 2) and recombination zone position (Supplementary Figure 4) with background bias support this notion. Although it is difficult to quantify how big a role changes in charge balance may play in the EQE variation of Fig. 2, the fact that the EQE generally improves more for a given $V_{bias}$ than the associated decrease in non-radiative rate (as in Fig. 3) suggests that the impact of charge balance is not negligible.

In the high current density regime, the EQE roll-off is dominated by a combination of Joule heating and charge imbalance. The EL transient data in Fig. 6b suggest that, at $J\sim150$ A cm$^{-2}$, roughly half of the EQE decrease from its peak can be attributed to Joule heating. Having ruled out any contribution from Auger loss at this current density in Fig. 8, we attribute the remaining EQE loss to charge imbalance. It is non-trivial to quantify the magnitude of this loss; however, it is clear from previous work on both organic and inorganic LEDs[31,32] that it can become a

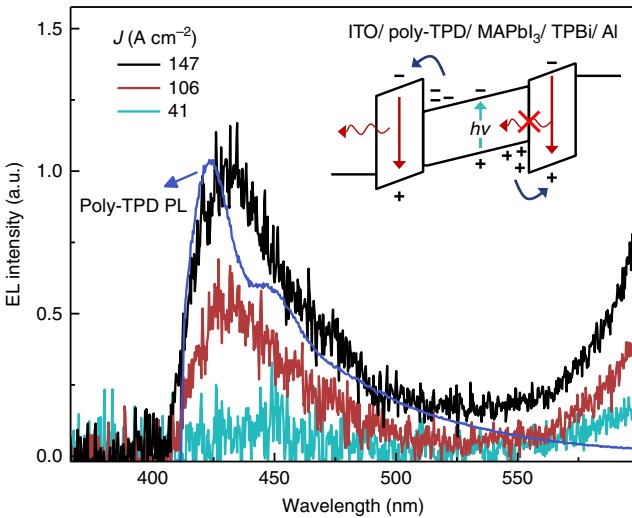

**Fig. 7** Evidence of charge imbalance. Electroluminescence spectra at high current density, demonstrating blue poly-TPD emission due to electron leakage out of the perovskite emissive layer as shown in the inset. Emission at wavelengths below 410 nm is cut off due to the use of a long-pass filter and the photoluminescence spectrum of poly-TPD (blue line) is overlaid for comparison

dominant factor due to the decreasing effectiveness of carrier confinement layers at high electric field and temperature (both factors strongly influence the rate of field-enhanced thermionic emission over confining energy barriers).

While the EQE and current density required to reach lasing threshold depend strongly on the details of the eventual resonator design, a rough estimate can be made based on the threshold carrier density, $n_{th} \sim 8 \times 10^{17}$ cm$^{-3}$, measured previously for MAPbI$_3$ in a metal-clad distributed feedback resonator (the most closely related existing structure to a laser diode) under optical pumping at $T = 160$ K[9]. Neglecting Auger losses, the corresponding threshold current density is approximately given by $J_{th} \approx q d_0 B n_{th}^2 / \eta_{IQE}$, where $q$ is the elementary charge, $d_0 \approx 100$ nm is the thickness of the perovskite active layer, $B \approx 4 \times 10^{-10}$ cm$^3$ s$^{-1}$ is the bimolecular radiative rate constant[23], and $\eta_{IQE}$ is the internal quantum efficiency of the diode[33]. Assuming a typical outcoupling efficiency of $\phi_{oc} = \eta_{EQE}/\eta_{IQE} \approx 0.15$ for the present LEDs[34], the corresponding EQE-current density product at threshold would be $\eta_{EQE} J_{th} \approx 62$ A cm$^{-2}$. By contrast, the EQE-current density product for the LEDs reported here peaks at 1.3 A cm$^{-2}$ (see Supplementary Figure 5), roughly 50 times lower than required. Weighting the needed improvement in favor of reducing EQE roll-off through improved charge balance rather than simply relying on higher current density is likely to be the most effective path forward since it both lowers Joule heating and reduces the likelihood that Auger will set in as a limiting factor.

In summary, we have investigated perovskite LEDs under short pulsed drive and found that their efficiency depends strongly on the manner in which they are driven. Factors such as duty cycle and background bias can change the EQE by more than an order of magnitude at a given current density, which is attributed to slow redistribution of ionic charge that changes both the internal field and non-radiative trap concentration in the device. This finding is closely related to the EQE hysteresis observed under DC drive and implies that the efficiency of many perovskite LEDs reported to date might be higher under optimal pulsed driving conditions.

Beyond efficiency, the driving scheme may also impact long-term operational degradation since holes that neutralize negatively charged iodide ions in the recombination zone (see Fig. 9b) will liberate neutral iodine that can subsequently diffuse into the adjacent electron transport layer and lead to decomposition of the

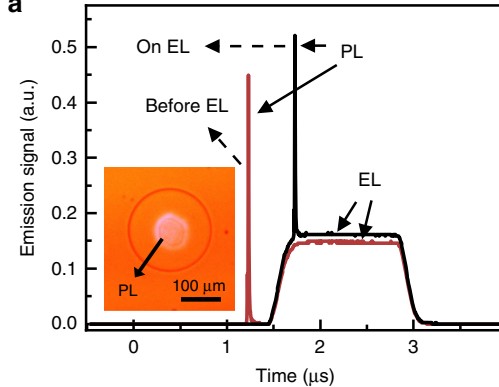

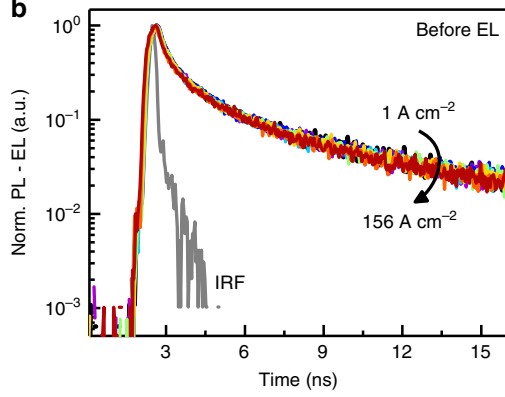

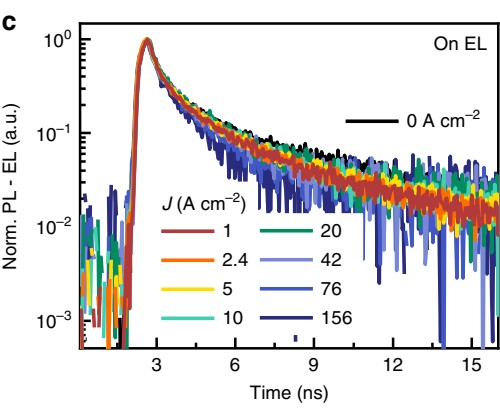

**Fig. 8** Relevance of Auger quenching. **a** Aggregate emission signal resulting from a 20 ps optical excitation pulse ($\lambda = 355$ nm, 10.7 μJ cm$^{-2}$ fluence) incident on an LED just before (red line) and just after the start of a 1.5 μs, 45 A cm$^{-2}$ electroluminescence pulse. The inset shows the overlap of the optical pump spot with the device area. The photoluminescence decay remains unchanged as a function of current density in both the before (**b**) and after (**c**) timing configurations up to the limit of 156 A cm$^{-2}$ reached in this experiment. Note that the faster overall decay of the photoluminescence transients here as compared to Fig. 3 results from the higher pump fluence used in this case. The instrument response function (IRF) is shown by the gray line with a temporal resolution of 0.2 ns

perovskite as discussed in ref. [35]. This hypothesis may help explain the large concentration of iodine found in the TPBi layer previously[36]. Better heat sinking is essential for operation at high current densities over 10 A cm$^{-2}$, even under short pulsed conditions. If Joule heating can be eliminated and improved transport layers can maintain balanced carrier injection into the emissive layer, it should enable efficient operation to be sustained at current densities over 100 A cm$^{-2}$. The fact that we observe no Auger loss in devices driven at this current density, just an order

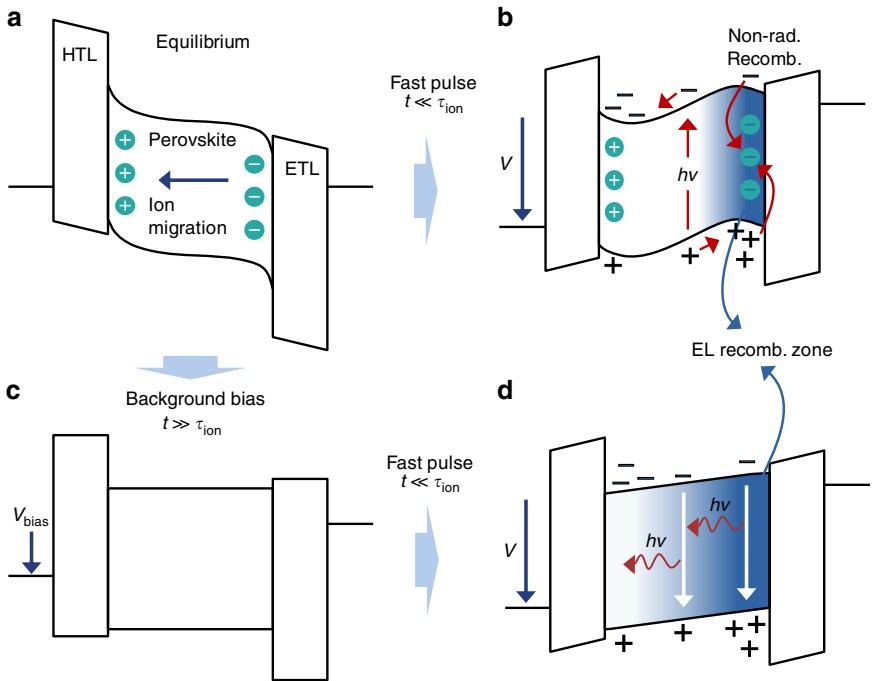

**Fig. 9** Model for pulsed operation. **a** At zero bias in thermal equilibrium, the work function difference between the contacts causes ions in the perovskite to drift, screening the internal field in the device. In this diagram, the ionic charge depicted results from the net displacement of positively and negatively charged ion populations and is not meant to imply drift of individual ions from one interface to the other. **b** Applying a sudden forward voltage pulse (faster than the characteristic timescale of ion motion, $\tau_{ion}$) tilts the overall band diagram accordingly, causing electrons and holes to accumulate near the emissive layer edges and recombine where the ionic defect concentration is highest. Photogenerated electrons and holes are swept to the emissive layer edges by the internal field and undergo similarly high non-radiative recombination. **c** Adding a small background bias can reverse the equilibrium ion drift, reducing the ion concentration near the edges of the perovskite and flattening its bands. **d** Applying a fast voltage pulse now leads to a more uniform distribution of injected electrons and holes and a lower concentration of ionic defects that act as non-radiative trap states in the recombination zone

of magnitude below that expected to enable electrically pumped lasing, suggests that the remaining barriers to a perovskite laser diode are technical rather than fundamental.

## Methods

**Fabrication**. MAI (BAI) was synthesized by mixing methylamine (n-butylamine) (Sigma Aldrich) with HI (Sigma Aldrich) in a 1:1 molar ratio. The reaction was performed in an ice bath while stirring for 3 h. The solvent of the resulting solution was evaporated using a rotary evaporator. The MAI (BAI) was recrystallized from an isopropyl alcohol/toluene mixture, filtered, and dried under low heat. Recrystallization, filtration, and drying were performed inside a $N_2$-filled glovebox. $PbI_2$ and MAI were dissolved in dimethylformamide (Sigma Aldrich, 99.8% anhydrous) to obtain a 0.4 M $MAPbI_3$ precursor solution. BAI was mixed with the $MAPbI_3$ precursor solution in a 0.2:1 molar ratio. Poly-TPD (6 mg ml$^{-1}$ in chlorobenzene) was spin coated on glass substrates with pre-patterned ITO and $SiO_2$ at 1500 rpm for 70 s followed by thermal annealing at 150 °C for 20 min. Poly-TPD was then treated with $O_2$ plasma for 6 s to improve wetting. Perovskite films were deposited on poly-TPD by spin coating at 6000 rpm. A solvent exchange step was performed after 3.5 s by dropping toluene on the spinning samples. Then, the samples were annealed at 70 °C for 5 min. The TPBi, LiF, and Al layers were thermally evaporated with thicknesses of 40 nm, 1.2 nm and 100 nm, respectively. All samples were encapsulated with encapsulation epoxy (Ossila) in conjunction with glass coverslips in a $N_2$-filled glovebox before transferring out for electrical and optical characterization.

To pattern $SiO_2$ on ITO, photoresist (AZ5214E) was spin-coated at 4000 rpm for 40 s on an ITO/glass substrate. After baking the photoresist layer at 100 °C for 70 s, image-reversal photolithography was carried out, followed by development for 90 s. An insulating 150 nm-thick $SiO_2$ layer was grown by plasma enhanced chemical vapor deposition at 90 °C. The small-area LED pattern was finally obtained by lift-off and then soaking the sample in photoresist remover (AZ1165) at 80 °C overnight.

**Characterization**. A voltage pulse train generated from a digital delay generator (Stanford Research System, DG645) was amplified by a custom designed amplifier and applied to the LEDs with background bias provided by a series-connected DC power supply. Current was determined by measuring the voltage across a 10 Ohm termination resistor amplified by a high-speed amplifier and detected using an oscilloscope. Electroluminescence was measured using a calibrated Si photodetector

and a fast photodiode connected to a wide bandwidth transimpedance amplifier for DC and pulsed measurements, respectively. In situations where Joule heating leads to changes in the current and EL intensity during the pulse (as in Fig. 6), the EQE was determined from the average value of each in the final 250 ns. The measurements reported in this work were collected over a 6-month period from encapsulated devices fabricated in four separate fabrication runs. We observe no environmental degradation over the course of 1 month and the EQE measured from run to run and device to device typically varies by less than 10% under pulsed drive. High current pulsed JVL sweeps (forward and backward) were acquired in less than 2 min to minimize degradation above 100 A cm$^{-2}$ per Supplementary Figure 2. Electroluminescence imaging was performed with an inverted microscope and an electron multiplying CCD camera. Background illumination was provided by a $\lambda$=660 nm laser diode using a series of neutral density filters to tune the power.

Spectrally resolved transient PL and EL spectra were collected using a Hamamatsu C10910 streak camera (10 ps temporal resolution) with a monochromator using an optical parametric oscillator (20 ps pulse width and 1 kHz repetition rate) for excitation. In cases where PL was collected during an EL pulse, the baseline EL signal was subtracted from the PL transient. Steady-state spectra were measured through an inverted microscope ported to a spectrograph with a cooled Si CCD array. Electroabsorption spectra were recorded using monochromatic light from a Xe lamp incident at approximately 15° through the ITO anode and detected in reflection from the metal cathode (i.e., a double pass in the perovskite layer) with a Si photodetector and current pre-amplifier. A sinusoidal dither was superimposed on the LED bias and the reflection signal was detected synchronously using a lock-in amplifier. Transient EA was subsequently carried out by fixing the incident wavelength to $\lambda = 730$ nm and using an oscilloscope to monitor the EA signal resulting from a $-2$ V to 2 V square wave with periods ranging from 10 ms and 2 s.

**Analysis**. PL transients were fit with an ABC rate model:[23] $dn/dt = -An - Bn^2 - Cn^3$, where $n$ is the charge carrier density, $t$ is time, A is the trap-assisted recombination coefficient, B is the radiative bimolecular rate constant, and C is the Auger rate constant. The fits in Fig. 3 were performed with a nonlinear least squares regression neglecting the Auger C coefficient due to the low pump fluences (0.2 μJ cm$^{-2}$) used for excitation. Transient thermal modeling was carried out in Comsol Multiphysics$^{TM}$ using the heat transfer module. Uniform volumetric heat generation was assumed in the perovskite layer according to the dissipated electrical power. The respective thermal conductivity and heat capacity assumed for each layer are $\kappa = 0.30$ W m$^{-1}$ K$^{-1}$ and $C_p = 241.9$ J

$kg^{-1}K^{-1}$ for MAPbI$_3$, $\kappa = 1.38$ W m$^{-1}$K$^{-1}$ and $C_p = 703$ J kg$^{-1}$K$^{-1}$ for SiO$_2$, and $\kappa = 238$ W m$^{-1}$K$^{-1}$ and $C_p = 900$ J kg$^{-1}$K$^{-1}$ for aluminum.

## Data availability

The data that support the findings of this study are available from the corresponding author upon reasonable request.

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

## Acknowledgements

This work was supported in part by the Air Force Office of Scientific Research under Award no. FA9550-18-1-0037 and by the National Science Foundation under Grant No. DMR-1654077.

## Author contributions

K.R. and L.Z. fabricated the devices, A.J.G. designed the pulse driver circuit, and H.K., J.S.P. and M.L. characterized the device performance with assistance from A.N.B. in measuring electroabsorption. H.K. conducted the spectroscopic measurements and carried out the data analysis. B.P.R and N.C.G supervised the work. All authors contributed in writing the manuscript.

## Additional information

**Competing interests:** The authors declare no competing interests.

