## [Peer Review File · Nature Communications]

Reviewers' comments:

Reviewer #1 (Remarks to the Author):

The high current injection into perovskite layers is a hot topic and valuable insights are provided by this paper. However, two points should be clarified to improve the manuscript.

In Fig. 4, while authors assigned the blue emission to poly-TPD, it is susceptible. Some previous reports show the PL peak at around 450 nm which is inconsistent with the obtained EL spectrum. Authors should show the PL of their poly-TPD film and confirm the consistency. Also, TPBi PL spectrum should be compared to exclude the possibility of carrier recombination at the opposite side. Basically poly-TPD does not accept electron injection in most multilayer structures of OLEDs. In some cases, electromer or excimer EL has been widely observed. Authors should consider the possibility of these excitonic states for detailed consideration.

Over all, the obtained EQE under short pulse operation of 2 micro-sec at high current of 100-1000 A/cm² region is less than 1%, that seems quite low value. Even we can escape Auger effect, what makes the EQE so low? It is unclear to explain such low EQE. When the lasing threshold is around ~10 micro J/cm², it seems far from lasing with the low EQE. It is required more accurate outlook for current driven lasing.

Reviewer #2 (Remarks to the Author):

This manuscript is devoted to engineering aspects of high-current-density LEDs based on hybrid perovskites. These studies are relevant to ongoing efforts on the realization of solution-processible perovskite laser diodes. The authors apply a fairly standard LED architecture comprised of organic electron and hole transport layers and further limit the injection area to 200 μm using a perforated SiO₂ spacer. This geometry allows them to reach very high current densities of ~600 A/cm². The authors use these devices to investigate the regimes of steady state and pulsed electrical excitation as well as the effect of a constant electrical or optical bias. The conclusion of this work is that the main EQE limiting factors are ion migration and Joule heating, while the role of Auger effects is insignificant.

This is a very detailed experimental study with a vast amount of factual information documented in 9 main figures and 3 SI figures. The fundamental science component of this work, however, is fairly weak. This manuscript does not report any breakthroughs or novel physical insights and reads like a collection of a laboratory notes. It is unlikely to be accessible by a general reader of Nat. Comm. Perhaps, after reorganization of materials and revisions it might become suitable for an applied, more specialized journal such as Adv. Opt. Mat., APL, JAP, IEEE J., etc.

A few general recommendations which might make this work more appealing to a general reader not familiar with a specific field of perovskite LEDs. The main motivation for this work seems to be the development of a device, which could reach optical gain and potentially lasing regimes. However, nowhere in the paper the authors discuss these regimes in quantitative terms. What is the critical carrier density required to obtain optical gain in these materials? What are the requirements for attaining ASE or a true lasing action? At the very end of the paper (literally, the last sentence before conclusions), the authors mention ~2 kA/cm² as a threshold requirement for a specific case of a "metal grating cavity." This is clearly not enough. The authors should estimate the gain and lasing thresholds in terms of carrier densities and then based on these values back out the corresponding

current densities. Then, they can use these estimations for benchmarking their experimental observations.

It would also be useful to include characterization of gain/lasing properties of perovskite layers using optical excitation. What are, for example, cw optical powers required to excite gain and ASE in these materials? Such data would help analyze EL measurements and, in particular, evaluate how close the authors are to realizing optical gain in their LEDs.

Minor notes:

Use of "luminance" is not appropriate for characterizing EL intensity of these devices. The EL peak is in the near-IR, i.e., outside a human eye visibility window. Standard light intensity units (e.g., W/cm²) are more appropriate.

Figure numbering should be fixed. Figures 6-8 are mislabeled.

To summarize, while this manuscript is a very detailed experimental study, it's not sufficiently appealing and impactful for a high profile journal such as Nat Comm. After appropriate revisions it might become suitable for one of the applied science journals.

Reviewer #3 (Remarks to the Author):

Kim and co-workers have presented a study on the operation of perovskite LEDs under intense electrical excitations. The manuscript is of potential interest to the researchers in the emerging field of perovskite LEDs, and its content falls within the scope of Nature Communications. The authors have carried out experiments to investigate the issue of EQE roll-off under current densities that are approximately an order of magnitude below the lasing threshold for typical hybrid perovskite materials. The paper attempts some of the interesting problems, including whether reliable LED operation can be sustained under very high current densities relevant to electrically-pumped lasers. Before I can recommend publication of this paper, the authors are advised to address the following issues.

1) It is true that high current density operation of perovskite LEDs is relatively unexplored and is a useful topic to be studied. But it is well known (and is pointed out by the authors) that the LEDs may experience irreversible degradation under high current densities. This degradation process may affect the reliability of analysis of the recombination process. For example, the data recorded for the same LED shown in Fig. 2b could be history dependent, and the order of the measurements may influence the results obtained. Have the authors investigated this issue – i.e., what was the sequence of the measurements and to what extent the subsequent measurements were affected by the earlier measurements? The authors are encouraged to perform LED degradation tests under at least one of the driving conditions investigated.

2) It is reported in the manuscript that the peak EQE under an optimum driving condition (short voltage pulses and a background bias) is substantially higher than the peak EQE obtained under standard steady-state conditions. However, in Fig. 2b, it is clear that the EQEs collected under DC currents of <30 mA/cm² are generally higher than the EQEs under voltage pulses and background bias. This 'low current density regime' is more relevant to display and low-intensity lighting purposes. So according to these results, it might be reasonable to say that the standard DC driving condition is more beneficial for achieving higher efficiencies for display or lighting applications. The authors should explain this observation (that pulsed electrical excitations could not achieve similar EQEs under low currents) more clearly or comment on this matter in greater detail.

3) The conclusion of Auger recombination being irrelevant at the very high current densities tested is encouraging. However, the peak EQEs at high currents are about one order of magnitude lower than

that of the peak EQEs under optimum conditions (likely due to much-reduced charge balance), and the relative voltage (and current) differences introduced by the background bias are smaller under high excitation conditions; so it might be possible to argue that the insignificant dependence of PL/EL yields on current densities is not a surprise. While this is not to question the validity of the statement that no evidence for the Auger process is found from these measurements, the authors are advised to clarify this issue further.

4) The methods section states that The PL and EL spectra were collected using a streak camera/monochromator/UV pump pulse set-up with a 10-20 ps time resolution. But the step size of the PL measurements was not specified. According to Figure 3, the initial rise of the PL signal after the excitation has a characteristic time of about 1 ns. If the temporal resolution of the setup is 10-20 ps, this PL rise has to be related to the emission behavior of the perovskite itself after the excitation. The 1 ns initial rise of PL should be explained.

5) The authors used an established ABC model to study the recombination processes in the perovskite using a combination of PL and EL. However, the experiment used to extract the trap-mediated recombination constant (A) is based on the transient PL data with a maximum time window of only about 50 ns. On this timescale, the PL is likely dominated by the radiative bimolecular process (even when the pulse energy is low) and the time window is therefore insufficient for the purpose of extracting constant A described in the ABC model. Ideally, the time window should be extended to microseconds to allow a more reliable measurement/fitting.

6) On the analysis and discussion of diode ideality factors; Figure 4c shows ideality factor values of below 1 and well above 2. These values cannot be interpreted using standard diode models and thus should be explained.

7) The authors should comment on the reliability of the EQE measurements or to provide more details about how the EQE values were obtained, especially for measurements that were not performed under standard DC conditions.

We appreciate the informative feedback and constructive criticism from the reviewers. We have addressed each comment/question by the reviewers as detailed below. Original referee comments are listed in *italics*, our responses are in black, and revisions in the manuscript are highlighted in yellow.

Response to Referee #1:

Overall, the obtained EQE under short pulse operation of 2 micro-sec at high current of 100-1000 A/cm² region is less than 1%, that seems quite low value. Even we can escape Auger effect, what makes the EQE so low? It is unclear to explain such low EQE. When the lasing threshold is around ~10 micro J/cm², it seems far from lasing with the low EQE. It is required more accurate outlook for current driven lasing.

The low EQE at high current is due to a combination of Joule heating and charge imbalance. Figure 6 quantifies the reduction due to Joule heating where, at a current density of 150 A cm⁻², the EL intensity decreases to roughly 60% of its starting value by the end of the pulse (the current density also increases slightly), resulting in a commensurate EQE reduction. Having ruled out any contribution from Auger loss at this current density in Fig. 8, we attribute the remaining EQE loss to charge imbalance (i.e. current leakage from the emissive layer). The transport layer emission spectrum in Fig. 7 proves the existence of charge imbalance; however, it is non-trivial to quantify the magnitude of this loss, as attested to by the large volume of literature studying EQE droop in inorganic LEDs. [For a review, see *Laser Photonics Rev.* 7, 3, 408 (2013)] Nonetheless, it is clear from both the inorganic and organic LED literature [*Laser Photonics Rev.* 7, 3, 408 (2013), *Physical Review B*, 77, 235215 (2008)] that charge imbalance can dominate EQE droop due to the decreasing effectiveness of carrier confinement layers at high electric field and temperature (both factors exponentially increase the rate of field-enhanced thermionic emission over confining energy barriers).

While the EQE and current density required to reach lasing threshold depends strongly on the details of the eventual resonator design, we can provide a rough estimate based on the threshold carrier density that we have previously measured for MAPbI₃ in a metal-clad distributed feedback resonator (the most closely related existing structure to a laser diode detailed in *Nano Lett.* **16**, 4624–4629 (2016)) under optical pumping, which is $n_{\text{th}} \sim 8 \times 10^{17} \text{ cm}^{-3}$ at a temperature of 160 K. Neglecting Auger losses, the corresponding threshold current density is approximately given by $J_{\text{th}} \approx qd_0Bn_{\text{th}}^2/\eta_{\text{IQE}}$, where q is the elementary charge, $d_0 \approx 100 \text{ nm}$ is the thickness of the perovskite active layer, $B \approx 4 \times 10^{-10} \text{ cm}^3\text{s}^{-1}$ is the bimolecular radiative rate constant at 160 K, and η_{IQE} is the internal quantum efficiency of the diode [L. A. Coldren and S.W. Corzine, *Diode Laser and Photonics Integrated Circuits*, John Wiley & Sons, (1995)]. Assuming a typical outcoupling efficiency of $\phi_{\text{oc}} = \eta_{\text{EQE}}/\eta_{\text{IQE}} \approx 0.15$ for the present LEDs, (*Adv. Optical Mater.*, **6**, 1800667, (2018)) the corresponding EQE-current density product at threshold

would be $\eta_{\text{EQE}}J_{\text{th}} \approx 62 \text{ A cm}^{-2}$. By contrast, the highest EQE-current density product achieved from the data in Fig. 2b is 1.3 A cm^{-2} , roughly 50 times lower than required. Weighting the needed improvement in favor of reducing EQE roll-off through improved charge balance rather than simply relying on higher current density is likely to be the most effective path forward since it both lowers Joule heating and reduces the likelihood that Auger will set in as a limiting factor.

To clarify the above points in the manuscript, we have added the following text on page 10 to conclude the Discussion section:

“In the high current density regime, the EQE roll-off is dominated by a combination of Joule heating and charge imbalance. The EL transient data in Fig. 6b suggest that, at $J \sim 150 \text{ A cm}^{-2}$, roughly half of the EQE decrease from its peak can be attributed to Joule heating. Having ruled out any contribution from Auger loss at this current density in Fig. 8, we attribute the remaining EQE loss to charge imbalance. It is non-trivial to quantify the magnitude of this loss; however, it is clear from previous work on both organic and inorganic LEDs^{30,31} that it can become a dominant factor due to the decreasing effectiveness of carrier confinement layers at high electric field and temperature (both factors strongly influence the rate of field-enhanced thermionic emission over confining energy barriers).

While the EQE and current density required to reach lasing threshold depend strongly on the details of the eventual resonator design, a rough estimate can be made based on the threshold carrier density, $n_{\text{th}} \sim 8 \times 10^{17} \text{ cm}^{-3}$, measured previously for MAPbI₃ in a metal-clad distributed feedback resonator (the most closely related existing structure to a laser diode) under optical pumping at T=160 K.⁸ Neglecting Auger losses, the corresponding threshold current density is approximately given by $J_{\text{th}} \approx qd_0Bn_{\text{th}}^2/\eta_{\text{IQE}}$, where q is the elementary charge, $d_0 \approx 100 \text{ nm}$ is the thickness of the perovskite active layer, $B \approx 4 \times 10^{-10} \text{ cm}^3\text{s}^{-1}$ is the bimolecular radiative rate constant²², and η_{IQE} is the internal quantum efficiency of the diode.³² Assuming a typical outcoupling efficiency of $\phi_{\text{oc}} = \eta_{\text{EQE}}/\eta_{\text{IQE}} \approx 0.15$ for the present LEDs,³³ the corresponding EQE-current density product at threshold would be $\eta_{\text{EQE}}J_{\text{th}} \approx 62 \text{ A cm}^{-2}$. By contrast, the EQE-current density product for the LEDs reported here peaks at 1.3 A cm^{-2} (see Supplementary Fig. 5), roughly 50 times lower than required. Weighting the needed improvement in favor of reducing EQE roll-off through improved charge balance rather than simply relying on higher current density is likely to be the most effective path forward since it both lowers Joule heating and reduces the likelihood that Auger will set in as a limiting factor.”

Figure S5 | EQE-current density product of the data in Fig. 2b shown relative to the threshold of $\eta_{\text{EQE}} J_{\text{th}} \approx 62 \text{ A cm}^{-2}$ estimated for a metal-clad distributed feedback laser diode operating at 160 K based on the results of Ref. [8].

In Fig. 4, while authors assigned the blue emission to poly-TPD, it is susceptible. Some previous reports show the PL peak at around 450 nm which is inconsistent with the obtained EL spectrum. Authors should show the PL of their poly-TPD film and confirm the consistency. Also, TPBi PL spectrum should be compared to exclude the possibility of carrier recombination at the opposite side. Basically poly-TPD does not accept electron injection in most multilayer structures of OLEDs. In some cases, electromer or electroplex EL has been widely observed. Authors should consider the possibility of these excitonic states for detailed consideration.

We have superimposed the PL spectrum of poly-TPD onto the EL spectra from our original Figure 7 below to highlight the close resemblance between the two.

Because these spectra were originally acquired with a 400 nm long-pass filter to avoid a grating double at longer wavelengths, we also went back and remeasured the LED emission without the long-pass filter to check for possible TPBi emission in the ultraviolet region. This is shown together with TPBi PL in the figure below, which confirms that no TPBi emission is observable in the EL spectrum. As noted in the manuscript, this does not mean that hole leakage into TPBi is insignificant, but rather that we simply are not sensitive to it since any TPBi emission would be absorbed by MAPbI₃ before exiting the device. Emission from an electromer or electroplex is similarly ruled out since these species typically emit at lower energies than the excitonic emission of poly-TPD or TPBi, which is not observed here.

To address this point in the manuscript, we have added the poly-TPD PL spectrum to Fig. 7 and revised the text on page 9 to read:

“This emission emerges above $J \sim 50 \text{ A cm}^{-2}$ and strongly resembles poly-TPD photoluminescence (overlaid for comparison), indicating the presence of electron leakage into (and subsequent EL from) the hole transport layer. Hole leakage into the TPBi electron transport layer may also occur; however, any TPBi emission would likely go undetected due to strong absorption by the perovskite layer as illustrated in the inset diagram.”

Response to Referee #2:

A few general recommendations which might make this work more appealing to a general reader not familiar with a specific field of perovskite LEDs. The main motivation for this work seems to be the development of a device, which could reach optical gain and potentially lasing regimes. However, nowhere in the paper the authors discuss these regimes in quantitative terms. What is the critical carrier density required to obtain optical gain in these materials? What are the requirements for attaining ASE or a true lasing action? At the very end of the paper (literally, the last sentence before conclusions), the authors mention $\sim 2 \text{ kA/cm}^2$ as a threshold requirement for a specific case of a “metal grating cavity.” This is clearly not enough. The authors should estimate the gain and lasing thresholds in terms of carrier densities and then based on these values back out the corresponding current densities. Then, they can use these estimations for benchmarking their experimental observations.

It would also be useful to include characterization of gain/lasing properties of perovskite layers using optical excitation. What are, for example, cw optical powers required to excite gain and ASE in these materials? Such data would help analyze EL measurements and, in particular, evaluate how close the authors are to realizing optical gain in their LEDs.

The requested information was determined previously in Refs. 8 and 10, where we investigated optically-pumped lasing from the same MAPbI₃ gain medium under pulsed and continuous-wave excitation, respectively. Reference 8 in particular explores a metal-clad distributed feedback MAPbI₃ laser structure, which is the most closely related structure to a potential laser diode that has been experimentally measured to date. The data from this work provides the basis for our expanded discussion of LED performance in the context of lasing threshold that we have now added to the Discussion section on page 11 of the manuscript together with an additional Supplementary Figure:

“While the EQE and current density required to reach lasing threshold depends strongly on the details of the eventual resonator design, a rough estimate can be made based on the threshold carrier density, $n_{\text{th}} \sim 8 \times 10^{17} \text{ cm}^{-3}$, measured previously for MAPbI₃ in a metal-clad distributed feedback resonator (the most closely related existing structure to a laser diode) under optical pumping at $T=160 \text{ K}$.⁸ Neglecting Auger losses, the corresponding threshold current density is approximately given by $J_{\text{th}} \approx qd_0 B n_{\text{th}}^2 / \eta_{\text{IQE}}$, where q is the elementary charge, $d_0 \approx 100 \text{ nm}$ is the thickness of the perovskite active layer, $B \approx 4 \times 10^{-10} \text{ cm}^3 \text{ s}^{-1}$ is the bimolecular radiative rate constant²², and η_{IQE} is the internal quantum efficiency of the diode.³² Assuming a typical outcoupling efficiency of $\phi_{\text{oc}} = \eta_{\text{EQE}} / \eta_{\text{IQE}} \approx 0.15$ for the

present LEDs,³³ the corresponding EQE-current density product at threshold would be $\eta_{\text{EQE}}J_{\text{th}} \approx 62 \text{ A cm}^{-2}$. By contrast, the EQE-current density product for the LEDs reported here peaks at 1.3 A cm^{-2} (see Supplementary Fig. 5), roughly 50 times lower than required. Weighting the needed improvement in favor of reducing EQE roll-off through improved charge balance rather than simply relying on higher current density is likely to be the most effective path forward since it both lowers Joule heating and reduces the likelihood that Auger will set in as a limiting factor.”

Figure S5 | EQE-current density product of the data in Fig. 2b shown relative to the threshold of $\eta_{\text{EQE}}J_{\text{th}} \approx 62 \text{ A cm}^{-2}$ estimated for a metal-clad distributed feedback laser diode operating at 160 K based on the results of Ref. [8].

Use of “luminance” is not appropriate for characterizing EL intensity of these devices. The EL peak is in the near-IR, i.e., outside a human eye visibility window. Standard light intensity units (e.g., W/cm²) are more appropriate.

Figure numbering should be fixed. Figures 6-8 are mislabeled.

The figure numbering problem has been corrected (the mislabeling appears to have been a pdf conversion problem) and the EL scale in Fig. 2a and 6b has been changed to units of irradiance as suggested.

Figure 2 | Operation under DC and pulsed drive. a, Current density (left-hand axis) and irradiance (right-hand axis) versus voltage for a 200 μm -diameter device operated in DC (black lines) and pulsed (colored lines) drive; ...

Figure 6 | Joule heating at high current density. Transient (a) current density and (b) irradiance at different pulsed voltages.

Response to Referee #3:

1) It is true that high current density operation of perovskite LEDs is relatively unexplored and is a useful topic to be studied. But it is well known (and is pointed out by the authors) that the LEDs may experience irreversible degradation under high current densities. This degradation process may affect the reliability of analysis of the recombination process. For example, the data recorded for the same LED shown in Fig. 2b could be history dependent, and the order of the measurements may influence the results obtained. Have the authors investigated this issue – i.e., what was the sequence of the measurements and to what extent the subsequent measurements were affected by the earlier measurements? The authors are encouraged to perform LED degradation tests under at least one of the driving conditions investigated.

We spent significant effort early on in our experiments identifying the conditions under which degradation becomes a factor and have bracketed all of our measurements by sweeping up and then back down in current density to check for this possibility in each measurement. In general, we find insignificant degradation as long as the current density is maintained below $\sim 100 \text{ A cm}^{-2}$ for $2 \mu\text{s}$ pulses, whereas degradation becomes evident at higher currents (this follows from Fig. 2b, where the forward and backward sweeps no longer fully retrace each other once the maximum current exceeds $\sim 100 \text{ A cm}^{-2}$).

The figure below provides a more explicit measurement of degradation versus time at high current density. The conclusion from this plot is similar: degradation is largely insignificant when the current density is kept below $\sim 100 \text{ A cm}^{-2}$ but becomes more substantial above it. The reason that degradation accelerates strongly above 100 A cm^{-2} is probably related to Joule heating as evident from the simulations in Fig. 6d, where the temperature rise in these particular devices rapidly accelerates as the current density increases beyond 100 A cm^{-2} .

We have added this figure and associated discussion regarding degradation to the Supplementary Information and referenced it in the main text on page 4:

“Higher currents up to $J = 620 \text{ A cm}^{-2}$ (limited by the driving circuit) can be reached without catastrophic device failure; however, this leads to irreversible device degradation as evident from the return sweep of the brown curve in Fig. 2b. **In general, we find that degradation is insignificant over the time scale of our measurements for current densities below $\sim 100 \text{ A cm}^{-2}$; Supplementary Fig. 2 quantifies the rate of degradation observed at higher current densities.**”

The new Supplementary Fig. 2 and its associated caption are:

Figure S2 | Current density of 200 μm -diameter LEDs recorded as a function of time for continuous pulsed excitation with 2 μs pulses at 20 Hz repetition rate. The degradation is irreversible and appears to be associated with Joule heating, where the temperature rise during a 2 μs pulse grows rapidly as the current density increases beyond 100 A cm^{-2} according to Fig. 6d in the manuscript.

2) It is reported in the manuscript that the peak EQE under an optimum driving condition (short voltage pulses and a background bias) is substantially higher than the peak EQE obtained under standard steady-state conditions. However, in Fig. 2b, it is clear that the EQEs collected under DC currents of $<30 \text{ mA/cm}^2$ are generally higher than the EQEs under voltage pulses and background bias. This ‘low current density regime’ is more relevant to display and low-intensity lighting purposes. So according to these results, it might be reasonable to say that the standard DC driving condition is more beneficial for achieving higher efficiencies for display or lighting applications. The authors should explain this observation (that pulsed electrical excitations could not achieve similar EQEs under low currents) more clearly or comment on this matter in greater detail.

One of the key findings stressed in this work is that the pulsed EQE depends strongly on the form of the driving pulse train. While the lowest current pulsed EQE data (filled symbols in Fig. 2b, with 70 μs pulse width and 130 μs between pulses) clearly do fall below the DC EQE at current densities $<30 \text{ mA cm}^{-2}$ as noted by the reviewer, decreasing the duty cycle leads to a very different trend (open symbols, where the time between the same pulses is increased to 2 ms), where the pulsed EQE trends upward with decreasing current density and might well intersect with the DC EQE.

Unfortunately, signal-to-noise becomes a limiting factor for these short pulsed measurements at low current density (short pulses, low current density, and small device area require measuring very small absolute current and light signals with high bandwidth, which is inherently challenging from a signal recovery and amplification standpoint), preventing us from directly investigating what actually occurs in the low current regime. Nonetheless, in light of the point made above, we feel that our original statement on page 11 that ‘the efficiency of many perovskite LEDs reported to date might be higher under optimal pulsed driving conditions’ remains fair overall, and explicitly valid in the high current regime that is the focus of this work. This is not to diminish the point made by the referee above, but rather motivates a focused study of pulsed operation in the low current regime given its critical importance for display applications, where pixels turn on and off at video refresh rates exceeding 200 Hz.

3) The conclusion of Auger recombination being irrelevant at the very high current densities tested is encouraging. However, the peak EQEs at high currents are about one order of magnitude lower than that of the peak EQEs under optimum conditions (likely due to much-reduced charge balance), and the relative voltage (and current) differences introduced by the background bias are smaller under high excitation conditions; so it might be possible to argue that the insignificant dependence of PL/EL yields on current densities is not a surprise. While this is not to question the validity of the statement that no evidence for the Auger process is found from these measurements, the authors are advised to clarify this issue further.

We have clarified the origin of the low EQE at high current density by adding the following paragraph to the Discussion section on page 10 of the manuscript:

“In the high current density regime, the EQE roll-off is dominated by a combination of Joule heating and charge imbalance. The EL transient data in Fig. 6b suggest that, at $J \sim 150 \text{ A cm}^{-2}$, roughly half of the EQE decrease from its peak can be attributed to Joule heating. Having ruled out any contribution from Auger loss at this current density in Fig. 8, we attribute the remaining EQE loss to charge imbalance. It is non-trivial to quantify the magnitude of this loss; however, it is clear from previous work on both organic and inorganic LEDs^{30,31} that it can become a dominant factor due to the decreasing effectiveness of carrier confinement layers at high electric field and temperature (both factors strongly influence the rate of field-enhanced thermionic emission over confining energy barriers).”

4) The methods section states that The PL and EL spectra were collected using a streak camera/monochromator/UV pump pulse set-up with a 10-20 ps time resolution. But the step size of the PL measurements was not specified. According to Figure 3, the initial rise of the PL signal after the excitation has a characteristic time of about 1 ns. If the temporal resolution of the setup is 10-20 ps, this PL rise has to be related to the emission behavior of the perovskite itself after the excitation. The 1 ns initial rise of PL should be explained.

The temporal resolution of the streak data depends on the sweep time used in the measurement and is roughly 1/100 of the time window. Our maximum time resolution of 10 ps is realized in a 1 ns time

window; however, the time window employed in Fig. 3 was 50 ns and therefore its temporal resolution is 0.5 ns. We have added the instrument response function (IRF) in Figure 3 and also in 8 to clarify this point and to clearly show that the initial rise of the PL signal is set by the IRF in this measurement and not by any carrier dynamics intrinsic to the perovskite itself.

Figure 3 | Transient photoluminescence during voltage pulses. Transient photoluminescence decays acquired from an optical pump pulse ($\lambda = 355$ nm, ~ 20 ps pulse width, fluence = $0.2 \mu\text{J cm}^{-2}$) incident $1 \mu\text{s}$ before (P1) and after (P2) the start of a 0.05 A cm^{-2} electrical pulse with different background bias levels as depicted in the inset diagram. Solid black lines are fits to the ABC rate equation described in the text. **The instrument response function (IRF) is depicted by the gray line.**

Figure 8 | Relevance of Auger quenching. . . . Note that the faster overall decay of the photoluminescence transients here as compared to Fig. 3 results from the higher pump fluence used in this case. **The instrument response function (IRF) is shown by the gray line.**

5) The authors used an established ABC model to study the recombination processes in the perovskite using a combination of PL and EL. However, the experiment used to extract the trap-mediated recombination constant (A) is based on the transient PL data with a maximum time window of only about 50 ns. On this timescale, the PL is likely dominated by the radiative bimolecular process (even when the pulse energy is low) and the time window is therefore insufficient for the purpose of extracting constant A described in the ABC model. Ideally, the time window should be extended to microseconds to allow a more reliable measurement/fitting.

The form of the PL transients in Fig. 3 is sufficient to fully constrain the A and B terms in the ABC model (the excitation density is not high enough for any contribution from the C term) with relative standard errors < 5%. The figure below illustrates this visually: the data cannot be fit using the B term alone and clearly require inclusion of the A term for an accurate description.

While fitting data acquired over a longer time range would of course reduce the uncertainty in the extracted A value as the reviewer notes, it would come at the expense of increasing uncertainty in B. Because our goal was to be sensitive to both parameters in order to understand whether one or both change under different pulse/background bias conditions – and we found that B does indeed change as discussed on page 5 of the manuscript – we adjusted the time window and excitation fluence to provide the best possible compromise. Given that the uncertainty in the fitted A coefficients is much smaller than the difference between them, our primary conclusion that background bias reduces A by over a factor of two stands firm. In the revised manuscript, we have added the 95% confidence interval error bars associated with the A coefficients to communicate their uncertainty more clearly:

“Fitting these decay curves with a simple ABC rate model²² indicates that the background bias decreases the trap-related A coefficient from $(4.0 \pm 0.4) \times 10^7 \text{ s}^{-1}$ to $(1.5 \pm 0.3) \times 10^7 \text{ s}^{-1}$.”

6) On the analysis and discussion of diode ideality factors; Figure 4c shows ideality factor values of below 1 and well above 2. These values cannot be interpreted using standard diode models and thus should be explained.

Values of the apparent ideality factor inferred from the slope $\eta = \left(\frac{kT}{q} \frac{\partial \ln J}{\partial V} \right)^{-1}$ typically exhibit a spurious increase at low voltage and high voltage due to parasitic shunt and series resistance, respectively. Both lead to a reduction in the slope of the IV curve and thus to an artificial increase in η . The figure below illustrates this visually for an equivalent circuit diode model with series and parallel resistance included.

For this reason, the ideality factor associated with this derivative is typically based on the minimum of the curve (see, e.g. Ref. 24 from the manuscript), which is very nearly 2.0 and 1.6 in Fig. 4c (black and red symbols, respectively), consistent with standard diode theory. The sub-unity value of the EL ideality factor (cyan) at low voltage is also spurious and occurs at EL turn-on when the photocurrent in the detector rises above the dark current; this feature is also common in previous measurements (see, e.g. Ref. 24 again) and is disregarded accordingly.

7) The authors should comment on the reliability of the EQE measurements or to provide more details about how the EQE values were obtained, especially for measurements that were not performed under standard DC conditions.

The measurements reported in this work were collected over a 6-month period from encapsulated devices fabricated in 4 separate fabrication runs. We observe no environmental degradation over the course of 4 weeks and the EQE measured from run-to-run and device-to-device typically deviates by less than 10% under pulsed drive. We have added this information on repeatability as well as additional details of the pulsed measurement procedure to the Methods section on page 13 of the manuscript as shown below:

“A voltage pulse train generated from a digital delay generator (Stanford Research System, DG645) was amplified by a custom designed amplifier and applied to the LEDs with background bias provided by a series-connected DC power supply. Current was determined by measuring the voltage across a 10 Ohm termination resistor, amplified by a high-speed amplifier (SR) and detected using an oscilloscope. Electroluminescence (EL) was measured using a calibrated Si photodetector and a fast photodiode connected to a wide bandwidth transimpedance amplifier for DC and pulsed measurements, respectively. In situations where Joule heating leads to changes in the current and EL intensity during the pulse (as in Fig. 6), the EQE was determined from the average value of each in the final 250 ns. The measurements reported in this work were collected over a six-month period from encapsulated devices fabricated in four separate fabrication runs. We observe no environmental degradation over the course of one month and the EQE measured from run-to-run and device-to-device typically varies by less than 10% under the pulsed drive.”

Voluntary changes.

We have added an energy level diagram in Figure 1a to more clearly convey the device structure.

Figure 1 | Device architecture. a, Energy level diagram and schematic of the device architecture, which consists of ITO (150 nm)/ poly-TPD (25 nm)/ MAPbI₃:BAI (100:20, 70 nm)/ TPBi (40nm)/ LiF (1.2 nm)/ Al (100 nm). The active area is defined by patterning openings in the insulating SiO₂ (150 nm) layer shown in the bottom graphic.

REVIEWERS' COMMENTS:

Reviewer #1 (Remarks to the Author):

The additional description and figures clearly explained my questions and the manuscript is ready for the acceptance.

Reviewer #3 (Remarks to the Author):

In general, the authors have responded to my comments and questions satisfactorily. The results presented are useful for understanding the emission behavior of perovskite LEDs operating under very high electrical injection conditions. Realistic estimates of lasing threshold under electrical excitations have also been provided. In my opinion, these findings can be accepted for publication in Nature Communications. The paper may be improved further by considering the optional minor corrections listed below.

1) In response to the first question I raised previously, the authors state that 'In general, we find that degradation is insignificant over the time scale of our measurements for current densities below $\sim 100 \text{ A cm}^{-2}$ '. It is important to clarify what the timescales of the measurements presented in the main paper are. Presumably, the timescale for the majority of measurements is less than 8 minutes, when the relative performance degradation is around or less than 1% for currents below 100 A cm^{-2} , according to the new figure presented by the authors. If the measurement duration is significantly longer than 8 min, such assumption can be problematic.

2) Under point 4 of my previous sets of comments, the authors responded that: 'however, the time window employed in Fig. 3 was 50 ns and therefore its temporal resolution is 0.5 ns'. I think it is necessary to mention this under the figure caption and/or under methods, so that the readers are informed that the temporal resolution of the measurement is 0.5 ns (rather than 10 ps) for these figures (Fig. 3 and Fig. 8).

Reviewer #4 (Remarks to the Author):

In this manuscript, the authors have experimentally investigated perovskite LEDs under intense short electric pulses driven. Such kind of investigation is very important for achieving electrically-pumped perovskite lasing. The obtained results and conclusions are interesting. For example, they found that the EQE depends strongly on the time-averaged history of the pulse train because the slow ion movement (likely dominated by movement of iodide vacancies and interstitials) changes the internal field distribution and trap density in the device at low current ($< 10 \text{ A cm}^{-2}$), as shown in Figure 9. While the impact of ions is less pronounced in the high current density regime ($> 10 \text{ A cm}^{-2}$), where EQE roll-off is dominated by a combination of Joule heating and charge imbalance. The previous reviewers' comments are nicely addressed. I would like to recommend publication of this manuscript.

The following comments and suggestions are not to criticize but strengthen the manuscript.

1. Both of the I-V characteristics presented in Fig. 2a and 4b are from DC driving devices. The device structures are supposed to be the same. However, the deviation between these two curves is even more than two order of magnitude. Therefore, in Fig. 4b, the tiny difference between the two plots with and without laser illumination is not enough for supporting the augment of reduction of trap-assisted recombination. Furthermore, the MAPbI₃ LED should give a photovoltaic effect with illumination of 660 nm laser photons, which is the possible reason for the higher threshold voltage

observed with light illumination.

2. I agree with the authors that the EQE droop was partially caused by imbalanced charge carrier injection. However, more evidences are required. In Fig. 7 the emission of poly-TPD is probably caused by the extension of recombination zone or overflow of electron in high driving current level as the comparable high mobility of hole and electron in MAPbI₃ polycrystalline film. Thus this result cannot support the conclusion of charge imbalance.

3. In Figure 8, the authors excluded the Auger contribution by comparatively probing the time-resolved photoluminescence (TRPL) with and without electric driving. However, this method is highly questionable. Since the authors excited the device with 355 nm laser pulses. The poly-TPD (25nm) has very large absorption coefficient at this excitation wavelength (Pls. see Figure 2 in Displays Volume 38, July 2015, Pages 32-37). I suggest the authors to repeat these TRPL measurement with excitation light around 600nm.

4. The authors described the charge carrier dynamics in their perovskite with the ABC model. The ABC model is suitable for the charge carrier dynamics dominated by free electron-hole bimolecular recombination, which is true in three-dimensional metal-halide perovskite. However, the charge carrier dynamics is dominated by first order excitonic recombination in low-dimensional perovskite (Nature communications 8, 14558, 2017). The perovskite used in this manuscript is a mixture of MAPbI₃ (100) with BAI (20). According to previous publication (Nature Photonics volume11, pages108–115 (2017)), this mixture should be perovskite nanocrystals.

5. The authors missed a relevant earlier reference of perovskite lasing. Please include the following reference in the introduction - Nature materials 13 (5), 476, 2014.

We appreciate the informative feedback and constructive criticism from the reviewers. We have addressed each comment/question by the reviewers as detailed below. Original referee comments are listed in *italics*, our responses are in black, and revisions in the manuscript are highlighted in yellow.

Response to Referee #3:

1) In response to the first question I raised previously, the authors state that ‘In general, we find that degradation is insignificant over the time scale of our measurements for current densities below $\sim 100 \text{ A cm}^{-2}$ ’. It is important to clarify what the timescales of the measurements presented in the main paper are.

The pulsed measurements above 100 A cm^{-2} all took less than 2 minutes to acquire, including both forward and backward voltage sweeps. To clarify this in the manuscript, we have added this detail to the Methods section on page 14:

“We observe no environmental degradation over the course of one month and the EQE measured from run-to-run and device-to-device typically varies by less than 10% under the pulsed drive. High current pulsed JVL sweeps (forward and backward) were acquired in less than 2 minutes to minimize degradation above 100 A cm^{-2} per Supplementary Figure 2.”

2) Under point 4 of my previous sets of comments, the authors responded that: ‘however, the time window employed in Fig. 3 was 50 ns and therefore its temporal resolution is 0.5 ns’. I think it is necessary to mention this under the figure caption and/or under methods, so that the readers are informed that the temporal resolution of the measurement is 0.5 ns (rather than 10 ps) for these figures (Fig. 3 and Fig. 8).

We added the comments accordingly under the captions of Fig.3 and Fig. 8.

In Fig. 3 caption

“The instrument response function (IRF) is depicted by the gray line with a temporal resolution of 0.5 ns”

In Fig. 8 caption

“The instrument response function (IRF) is shown by the gray line with a temporal resolution of 0.2 ns”

Response to Referee #4:

1. Both of the I-V characteristics presented in Fig. 2a and 4b are from DC driving devices. The device structures are supposed to be the same. However, the deviation between these two curves is even more than two order of magnitude. Therefore, in Fig. 4b, the tiny difference between the two plots with and without laser illumination is not enough for supporting the augment of reduction of trap-assisted recombination. Furthermore, the MAPbI₃ LED should give a photovoltaic effect with illumination of 660 nm laser photons, which is the possible reason for the higher threshold voltage observed with light illumination.

The deviation being referred to is in the parasitic shunt/chemical capacitance current of the device before it has actually turned on; the on-state current densities between Fig. 2a and 4b are fully consistent. The low voltage shunt/chemical capacitance current has little to do with the intrinsic diode physics of the device being investigated in Fig. 4b and varies instead with extrinsic factors such as device area, sweep rate, and direction (as apparent from the fact that it varies by more than two orders of magnitude between the forward and reverse sweeps in the *same* measurement in Fig. 2a). The on-state diode current is, by contrast, a consistent quantity and the ideality difference highlighted in Fig. 4b is fully reproducible.

2. I agree with the authors that the EQE droop was partially caused by imbalanced charge carrier injection. However, more evidences are required. In Fig. 7 the emission of poly-TPD is probably caused by the extension of recombination zone or overflow of electron in high driving current level as the comparable high mobility of hole and electron in MAPbI₃ polycrystalline film. Thus this result cannot support the conclusion of charge imbalance.

Extension of the recombination zone and overflow of electrons at high driving current are two different ways of referring to the same phenomenon, which is more generally referred to as charge imbalance (defined as the incomplete recombination of electron and hole currents within the MAPbI emissive layer) that is discussed extensively on page 8 and page 11 in the manuscript.

3. In Figure 8, the authors excluded the Auger contribution by comparatively probing the time-resolved photoluminescence (TRPL) with and without electric driving. However, this method is highly questionable. Since the authors excited the device with 355 nm laser pulses. The poly-TPD (25nm) has very large absorption coefficient at this excitation wavelength (Pls. see Figure 2 in Displays Volume 38, July 2015, Pages 32-37). I suggest the authors to repeat these TRPL measurement with excitation light around 600nm.

The TRPL measurement was spectrally resolved using a streak camera to isolate MAPbI PL (637 – 813 nm wavelength range) from poly-TPD or TPBi PL (380 – 500 nm wavelength range). Thus, there is no contribution from transport layer emission in the transients analyzed in Fig. 8. We have clarified this in the Methods section on page 14:

“Spectrally-resolved transient PL and EL data were collected using a monochromator and a Hamamatsu C10910 streak camera (10 ps temporal resolution) with an optical parametric oscillator (~20 ps pulse width and 1 kHz repetition rate) as the excitation source.”

4. The authors described the charge carrier dynamics in their perovskite with the ABC model. The ABC model is suitable for the charge carrier dynamics dominated by free electron-hole bimolecular recombination, which is true in three-dimensional metal-halide perovskite. However, the charge carrier dynamics is dominated by first order excitonic recombination in low-dimensional perovskite (Nature communications 8, 14558, 2017). The perovskite used in this manuscript is a mixture of MAPbI₃ (100) with BAI (20). According to previous publication (Nature Photonics volume 11, pages 108–115 (2017)), this mixture should be perovskite nanocrystals.

Our previous publication referred to by the reviewer above (*Nat. Phot.*, 11, 108 (2017)) showed that the 100:20 MAPbI₃:BAI molar ratio used here forms 3D perovskite crystallites with dimensionality $n > 30$ that exhibit free carrier rather than excitonic recombination. It was only at a 100:100 ratio that the films become dominated by the 2D perovskite ($n = 2$) where excitonic recombination is significant.

5. The authors missed a relevant earlier reference of perovskite lasing. Please include the following reference in the introduction - Nature materials 13 (5), 476, 2014.

We have added this reference to our manuscript and changed the numbering accordingly.

4. Xing, G. et al. Low-temperature solution-processed wavelength-tunable perovskites for lasing. *Nat. Mater.* 13, 476–480 (2014).